# Mechanism Repositioning Based on Integrative Pharmacology: Anti-Inflammatory Effect of Safflower in Myocardial Ischemia–Reperfusion Injury

**DOI:** 10.3390/ijms24065313

**Published:** 2023-03-10

**Authors:** Feng Zhao, Hong Jiang, Tong Zhang, Hong Chen, Weijie Li, Xin Li, Ping Wang, Haiyu Xu

**Affiliations:** 1Institute of Chinese Materia Medica, China Academy of Chinese Medical Sciences, Beijing 100700, China; 2College of Traditional Chinese Medicine, Shenyang Pharmaceutical University, Shenyang 110016, China; 3Key Laboratory for Research and Evaluation of Traditional Chinese Medicine, National Medical Products Administration, China Academy of Chinese Medical Sciences, Beijing 100700, China

**Keywords:** safflower, integrative pharmacology, myocardial ischemia–reperfusion injury, UPLC-QTOF-MS/MS, LAD ligation, inflammation-related signaling pathways

## Abstract

Safflower (*Carthamus tinctorius*. L) possesses anti-tumor, anti-thrombotic, anti-oxidative, immunoregulatory, and cardio-cerebral protective effects. It is used clinically for the treatment of cardio-cerebrovascular disease in China. This study aimed to investigate the effects and mechanisms of action of safflower extract on myocardial ischemia–reperfusion (MIR) injury in a left anterior descending (LAD)-ligated model based on integrative pharmacology study and ultra-performance liquid chromatography–quadrupole time-of-flight-tandem mass spectrometer (UPLC-QTOF-MS/MS). Safflower (62.5, 125, 250 mg/kg) was administered immediately before reperfusion. Triphenyl tetrazolium chloride (TTC)/Evans blue, echocardiography, terminal deoxynucleotidyl transferase-mediated dUTP nick-end labeling (TUNEL) assay, lactate dehydrogenase (LDH) ability, and superoxide dismutase (SOD) levels were determined after 24 h of reperfusion. Chemical components were obtained using UPLC-QTOF-MS/MS. Gene ontology (GO) and Kyoto Encyclopedia of Genes and Genomes (KEGG) analyses were performed. Quantitative real-time polymerase chain reaction (qRT-PCR) and Western blotting were used to analyze mRNA and protein levels, respectively. Safflower dose-dependently reduced myocardial infarct size, improved cardiac function, decreased LDH levels, and increased SOD levels in C57/BL6 mice. A total of 11 key components and 31 hub targets were filtered based on the network analysis. Comprehensive analysis indicated that safflower alleviated inflammatory effects by downregulating the expression of NFκB1, IL-6, IL-1β, IL-18, TNFα, and MCP-1 and upregulating NFκBia, and markedly increased the expression of phosphorylated PI3K, AKT, PKC, and ERK/2, HIF1α, VEGFA, and BCL2, and decreased the level of BAX and phosphorylated p65. Safflower shows a significant cardioprotective effect by activating multiple inflammation-related signaling pathways, including the NFκB, HIF-1α, MAPK, TNF, and PI3K/AKT signaling pathways. These findings provide valuable insights into the clinical applications of safflower.

## 1. Introduction

Acute myocardial ischemia (AMI) is a leading cause of morbidity and mortality worldwide, and it arises due to the disruption of a vulnerable atherosclerotic plaque or erosion of the coronary artery endothelium in most cases [1]. Timely myocardial reperfusion through percutaneous coronary intervention (PCI) or thrombolysis is the most effective method for treating myocardial infarction injury. However, it is associated with serious adverse outcomes, including arrhythmias, reversible contractile dysfunction, endothelial and microvascular dysfunction, and lethal cell damage, which are collectively termed “myocardial ischemia–reperfusion (MIR) injury” [2,3]. MIR injury contributes to 50% of the final infarct size in AMI. Excessive inflammatory responses due to infiltration of circulating leukocytes following reperfusion aggravate cardiomyocyte death, resulting in the development of heart failure. Unfortunately, clinical trials have reported only few successful interventions for reperfusion injuries [4]. Therefore, the development of pharmacological interventions is a promising strategy for reperfusion injury therapy.

Traditional Chinese medicine (TCM) has more than 2000 years of history and widespread clinical applications. TCM therapies consider multi-ingredients and multi-targets, lead to few adverse reactions, and are cost-effective. TCM might be used as a complementary and alternative approach to the primary and secondary prevention of cardiovascular disease [5,6]. TCM syndromes of AMI refer to thoracic obstruction, which was considered as qi and blood deficiency, liver depression and spleen deficiency (transmural ischemia), blood stasis and obstruction collaterals, and qi stagnation and blood stasis (thrombotic occlusion) [7,8]. The traditional effects of safflower (*Carthamus tinctorius* L.) include the regulation of blood stasis and improvement of blood circulation. It is commonly used as herbal medicine and has been for more than 1000 years [9]. Modern pharmacological studies have demonstrated that safflower possesses anti-tumor, anti-thrombotic, anti-oxidative, immunoregulatory, and cardio-cerebral protective effects [10,11,12]. Clinical preparations of safflower, such as safflower injection and safflower yellow injection, have been used clinically for the treatment of cardio-cerebrovascular disease in China for many years [5,13,14,15]. As one of the important components of Danhong injection, it has been used in the clinical therapy of cardiovascular and cerebrovascular diseases in China for several years [10]. In addition, studies also indicated that safflower reduces MIR injury by increasing left ventricular systolic pressure (LVSP), rate of left ventricular pressure change (+dp/dtmax and -dp/dtmax), and by decreasing the levels of malonaldehyde [11,12]. However, there is little in-depth and comprehensive evidence on the mechanisms of safflower against MIR injury.

Traditional Chinese medicine integrative pharmacology (TCMIP) is an interdisciplinary subject based on the theory of TCM that comprehensively explores the interaction between various components of TCM and the body [16,17]. Briefly, by predicting the targets and pharmacological effects of herbal medicine, it enables the revelation of the association of the drug–gene–disease synergistic module, screens the synergistic multi-components, and clarifies the herbal ingredients and their related characteristics, as well as the relationship between the compound–target and target–disease [18]. This approach will be the next promising paradigm shift, from “one target, one component” to “network targets, multi-component” [19]. Ultraperformance liquid chromatography–quadrupole time-of-flight–tandem mass spectrometer (UPLC-QTOF-MS/MS) is a powerful tool for the qualitative characterization of chemical components in herbs by providing high chromatographic and mass resolution, accurate mass measurement, and abundant fragment ion information [20,21]. In this study, we used UPLC-QTOF-MS/MS for chromatographic separation and structural elucidation.

In the present study, we employed the Integrative Pharmacology-based Network Computational Research Platform of TCM (TCMIP v2.0, http://www.tcmip.cn/, accessed on 3 January 2022) to evaluate the efficacy of safflower in the treatment of MIR injury by using a classical mouse model of ligation of the left anterior descending (LAD) coronary artery, described the chemical fingerprint of safflower by UPLC-QTOF-MS/MS, explored the mechanism of action of safflower against MIR injury verified the key genes by qRT-PCR and Western blotting. The present study provides a comprehensive understanding of the multi-target regulation induced by safflower against MIR injury, and complements the discovery of key ingredients and targets (Figure 1).

## 2. Results

### 2.1. Effect of Safflower for Protection against MIR Injury in Mice

Mice were randomly divided into six groups, including the sham group, control group, safflower low-dose group, medium-dose group, high-dose group (62.5 mg/kg, 125 mg/kg, 250 mg/kg, 4 times/24 h, i.v.), and positive control metoprolol group (12.5 mg/kg/d, i.p.). Cardiac infarct size was determined by triphenyl tetrazolium chloride (TTC)/Evans blue staining in mice who had undergone 30 min ischemia and 24 h reperfusion (Figure 2A). Based on the similar area at risk (AAR) (Figure 2B), the control group had a higher infarct size than the other groups following MIR injury, suggesting that the MIR model was established successfully. Safflower treatment (62.5, 125, 250 mg/kg, 4 times/24 h, i.v.) and metoprolol treatment (12.5 mg/kg, once for 24 h, i.p.) significantly reduced the percentage of infarct area (IS) in AAR (Figure 2C). Cardiac function was analyzed using echocardiography 24 h after reperfusion (Figure 2F–L). As shown in Figure 2H,L, the left ventricular ejection fraction (EF%) and left ventricular shortening fraction (FS%) values in the control group were significantly lower than those in the sham group (*p* < 0.001). Safflower (62.5, 125, and 250 mg/kg, 4 times/24 h) significantly improved cardiac contractile function, as reflected by increasing in EF% and FS% (*p* < 0.05). Safflower (62.5, 125, and 250 mg/kg, 4 times/4 h) also improved the left ventricular end-diastolic anterior wall thickness (LVAWs), left ventricular internal diameter systolic (LVIDs), and left ventricular volume systolic (LV Volume s) (Figure 2J–L, *p* < 0.05). Safflower in the high-dose group (250 mg/kg, 4 times/24 h, i.v.) showed an effect equivalent to that observed in the positive control metoprolol group (12.5 mg/kg, once for 24 h, i.p.). Similarly, to verify the effect of safflower on myocardial necrosis and oxidative stress in the serum, we measured the levels of LDH and SOD (Figure 2D,E). We found that LDH activity in the control group was higher than that in the sham group, while the SOD activity was lower in the control group than that in the sham group. LDH activity in the groups treated with safflower profoundly reduced in a dose-dependent manner, whereas the level of SOD increased after treatment with safflower and metoprolol in MIR-injured mice. Furthermore, we determined cardiomyocyte death through TUNEL staining (Figure 2M,N). The TUNEL-positive cells increased in the control group. Safflower and metoprolol treatment markedly reduced apoptosis in myocardial tissue (Figure 2N). No apoptosis was observed in the sham group (Figure 2M).

### 2.2. Identification and Characterization of Chemical Constituents in Safflower

The base peak intensity (BPI) chromatograms of the water extraction of safflower in the positive and negative ion modes detected by UPLC-QTOF-MS/MS are shown in Figure 3. A total of 79 chemical compounds (51 in electrospray ionization [ESI^+^] and 51 in ESI^-^) in safflower were identified or tentatively characterized using UNIFI 1.8 software, and they included flavonoids, phenylpropanoids, alkaloids, lignans, and fatty acids. Among them, flavonoids accounted for 67% (Figure 3E) and glycosides accounted for 57% of the total phytoconstituents (Figure 3F). Detailed information on the chemical compounds is listed in Appendix A, including the component name, half relaxation time (t_R_, min), measured value (*m*/*z*), theoretical value (*m*/*z*), error (ppm), formula used, response value, and fragment. Hydroxysafflor yellow A (HSYA) and N1,N5,N10-(Z)-tri-p-cou-maroylspermidine were used as examples to illustrate the identification process in detail. As shown in Figure 3C, the ion [M + H]^+^ 613.17534 was mainly determined as the molecular ion peak of HSYA, and representative fragments were m/z 451.12319 [M + H-Glc]^+^, 433.1127 [M + H-Glc-OH]^+^, 415.10177 [M + H-C_6_H_14_O_7_]^+^,and 235.08432 [M + H-C_11_H_7_O_2_]^+^, all of which were identified as HSYA [22]. As For N_1_,N_5_,N_10_-(Z)-tri-p-cou-maroylspermidine [23], the most indicative fragments were identified at *m*/*z* 582.2624 [M-H]^-^, and the other fragments were 316.16630 [M-H-C_9_H_7_O_2_-C_8_H_7_O]^-^, 145.02862 [M-H-C_9_H_7_O_2_-C_8_H_7_O-C_8_H_17_ON_3_]^-^, and 119.04965 [M-H-C_9_H_7_O_2_-C_8_H_7_O-C_8_H_17_ON_3_-CO]^-^ (Figure 3D).

### 2.3. Target Prediction and Functional Enrichment

The TCMIP v2.0 database was used to search for candidate targets of safflower. A total of 486 major candidate human genes corresponding to 79 chemical compound identifications were collected with similarity scores of 0.7 (Appendix A). A total of 4579 MIR-related targets were obtained from three databases, including the GeneCard database (2341 genes), DisGeNET database (2173 genes), and TCMIP v2.0 database (65 genes), by searching for the keywords “myocardial ischemia–reperfusion injury,” “myocardial ischemia injury,” and “myocardial infarction injury.” A total of 859 targets were filtered for the next analysis with scores >10 for the GeneCard database, >0.1 for the DisGeNET database, and all the genes of the TCMIP v2.0 database.

To identify the common genes between safflower and MIR injury, a Venn diagram was developed, which screened 105 overlapping genes between MIR targets (859) and safflower-related genes (486) (Figure 4A). A total of 105 overlapping genes were subjected to Gene ontology (GO) and Kyoto Encyclopedia of Genes and Genomes (KEGG) functional enrichment analyses using the Database for Annotation, Visualization and Integrated Discovery (DAVID v6.8). The GO terms included 329 biological processes (BPs), 59 cellular components (CCs), and 72 molecular functions (MFs) (Appendix A, *p* < 0.05). The top 30 BP terms closely related to MIR injury are shown in Figure 4B, which suggests that biological processes mainly focus on the negative regulation of inflammation, hypoxia, and apoptosis. One hundred and forty-nine signaling pathways of safflower in MIR injury were enriched based on the KEGG database, 34 of which were optimally related to the corresponding pathological events involved in MIR injury (Figure 4C, Appendix A). These pathways could be divided into three functional modules: ① Inflammation, which includes the TNF signaling pathway, NFκB signaling pathway, natural killer cell-mediated cytotoxicity, T-cell receptor signaling pathway, Toll-like receptor signaling pathway, and NOD-like receptor signaling pathway. The inflammation module, compared with the others, was the primary module; ② Metabolism, which includes the AMPK signaling pathway, cAMP signaling pathway, and arachidonic acid metabolism. This module contributes to cell metabolism, ③ apoptosis, Ca^2+^ overloading, and oxidant stress.

### 2.4. Screening of Key Components and Hub Genes

To screen hub components and key targets, we constructed a multi-dimensional network using CytoScape 3.7. Based on the results of GO and KEGG analyses, 71 important targets were screened from the 34 pathological event pathways of MIR injury, which were regarded as effective targets in the therapeutic effect of safflower on MIR injury. The 71 important targets were discovered in 55 chemical compounds. Coincidentally, most of these were flavonoids, flavonoid glycosides, and other glycosides.

The 71 key targets, 55 active ingredients, and 34 KEGG pathways were entered into CytoScape 3.7, and a correlative network of “compounds-targets-pathways-symptoms” was constructed (Figure 5A). The active ingredients marked in blue were divided into three categories: flavonoids, flavonoid glycosides, and other compounds. Notably, flavonoids played a critical role in the treatment of MIR injury. The targets and pathways marked in orange and pink, respectively, were distributed into three groups based on pharmacological activities, including immune inflammation, metabolism, and others. In addition, gene classification was dependent on the frequency of gene targets in different pathways.

As shown in Figure 5A, immune inflammation accounted for the largest proportion. The interactive network consisted of 160 nodes and 729 edges, and detailed information is provided in Appendix A. The network was divided into three parts: components, targets, and pathways. The topological feature “degree” was chosen to identify topologically important nodes. Nodes with degree values greater than the median were filtered out as hub targets. For genes, the median value of “degree” was 6. Thirty-two nodes were selected as the hub genes (Table 1). The top ten genes were PRKCA, AKT1, CSNK2A1, CSNK2B, PIK3CG, HSPA2, PTGS1, MAPK1, HK1, and TNF.

The 31 hub genes were subjected to functional enrichment analysis again. The main functional module was significantly associated with immune inflammation and included the NFκB, PI3K/AKT, MAPK, HIF1α/VEGFA, TNF, and IL-17 signaling pathways (Figure 5B, Appendix A), which are marked in blue, and the size of the node represents its importance in the network. The five most significantly regulated pathways were chosen to be verified.

Homogenization is a challenging problem in the network pharmacology of TCM. To overcome the bottleneck problem, an integrative pharmacology strategy has been firstly applied for the effects of safflower on MIR injury by network analysis of Component–Target–Disease weighted by chemical composition content, literature mining based on key active components and bioactivities, and quantitative analysis [24,25,26,27,28,29,30,31,32,33,34,35,36,37,38,39,40]. A total of 19 chemical components with a degree greater than the median of 8 were selected. Literature mining was performed to verify the reliability of the key components prediction results, which were filtered by the median of 8. As shown in Appendix A, the literature based on animal studies or cell experiment with the model of hypoxia/reoxygenation or myocardial ischemia/reperfusion injury. In addition, the compound content was considered an important filter condition. As shown in Figure 3 and Appendix A, we listed the compounds in descending order of response value and screened the compounds with a median response value of 67667. A total of 11 compounds were filtered by the three above-mentioned factors, including quercetin (+40), luteolin (−44, +55), apigenin (−47, +58), rutin (+33), hydroxysafflor yellow A (−11, +17), kaempferol (−42, +44), baicalin (−40, +51), eriodictyol (+23), 6-hydroxyapigenin (−48, +46), 6-hydroxykaempferol (−32, +35), and 6-hydroxykaempferol 3-rutinoside-6-glucoside. Quercetin (+40), luteolin (−44, +55), apigenin (−47, +58), kaempferol (−42, +44), baicalin (−40, +51), and eriodictyol (+23) had higher degree values and higher response values. Rutin (+33) and HSYA (−11, +17) had a lot of literature evidence and higher response values. 6-hydroxyapigenin (−48, +46), 6-hydroxykaempferol (−32, +35), and 6-hydroxykaempferol 3-rutinoside-6-glucoside had high degree values and higher response values. However, they have not been reported thus far (Table 2).

### 2.5. The Mechanism and Molecular Docking

A diagram of the mechanism, including the hub targets, key components, and crucial signaling pathways, is shown in Figure 6A. Molecular docking was also performed to determine the binding affinities of key ingredients and hub targets. The crucial proteins, including PRKCA, PIK3CG, and AKT1, were filtered for molecular docking. The results showed that PRKCA, PIK3CG, and AKT1 had a good affinity with 11 hub compounds with stable conformations and high binding activity. As for PRKCA, a total of 10 of 11 hub compounds bind with binding affinities of less than -7.8 kcal/mol (Figure 6B, Appendix A), including 6-hydroxykaempferol-3-rutinoside-6-glucoside, 6-hydroxykaempferol, apigenin, baicalin, eriodictyol, HSYA, kaempferol, luteolin, quercetin, and rutin. And the five hub compounds, including eriodictyol, HSYA, kaempferol, luteolin, quercetin, and rutin, can bind to PIK3CG and AKT1 with the binding affinities of less than −5.9 (C) and −7.5 (D) kcal/mol, respectively, respectively (Figure 6C,D, Appendix A). In this study, the binding affinities of less than –5.0 kcal/mol were employed as screening criteria [97], and PyMOL 2.5 was used for visualization.

### 2.6. Safflower Inhibits Inflammation-Related Factors in MIR Mice

The accuracy of the network pharmacology prediction results was verified by performing qRT-PCR and Western blot. The NFκB, HIF-1α, MAPK, TNF, and PI3K/AKT signaling pathways are all involved in the activation of inflammation-related factors. The upstream hub genes were quantified through Western blotting, and the downstream inflammatory factors were detected by using qRT-PCR. Safflower treatment increased the expression of phosphorylation of PI3K and AKT, HIF1α, VEGFA, ERK1/2, and PKC, and decreased the level of phosphorylated p65 (Figure 7A–F). Safflower also inhibited the MIR-injured cardiomyocyte apoptosis, inhibited the expression of BAX, and promoted the expression of BCL2 (Figure 7G). As shown in Figure 7H–N, the expression of inflammation-related factors, including NFκB-1, IL-6, IL-1β, IL-18, MCP-1, and TNF-α, in the heart tissue of the mice in the control group was significantly upregulated compared to that in the sham group (*p* < 0.01), whereas the expression of NFκBia was downregulated. A dose-independent decrease in NFκB-1, IL-6, IL-1β, IL-18, MCP-1, and TNF-α expression was observed in safflower-treated mice (*p* < 0.05), and NFκBia mRNA expression significantly increased (*p* < 0.01). However, there were no significant differences in the mRNA levels of PI3K or HIF1α (Appendix A).

## 3. Discussion

In the present study, we identified 11 key components and 31 hub targets of safflower treatment in MIR injury using an integrative pharmacological strategy and focused on the mechanism of inflammatory response mediated by the NFκB, HIF-1α, MAPK, TNF, and PI3K/AKT signaling pathways. According to Network Pharmacology Evaluation Method Guidance, the network pharmacology evaluation is conducted from reliability, standardization, and rationality [98]. In our study, UPLC-QTOF-MS/MS, literature mining, and experimental verification ensured the reliability, and TCMIP v2.0 supplemented the standardization and rationality.

Inflammation plays a prominent role in MIR injury. During the initial stages of AMI, an acute inflammatory response is evoked, in which neutrophils infiltrate the myocardium via chemotactic attraction and aggravate the state of the already injured tissue. When neutrophils reach the reperfused tissue, they are exposed to chemotactic agents that are mainly released from endothelial cells and activated in their normal systemic circulation path [99]. This pro-inflammatory response is exacerbated and continues to cause cardiomyocyte death 6–24 h post-reperfusion [100]. Necrotic cardiac cells release nuclear factors, such as TLR4 and MCP-1, to activate the HIF-1α, MAPK, and PI3K/AKT signaling pathways, further promoting the NFκB signaling pathway [22,23,101]. Then the signaling pathways mediate the release of inflammatory cytokines, such as IL-6, IL-1β, IL-18, TNF-α, etc., leading to neutrophil attraction, sequestration, and adhesion [100]. Accompanied by the release of these cytokines, pro-inflammatory signaling pathways are activated again. Therefore, the inhibition of inflammatory factors during MIR injury effectively inhibits myocardial injury. β-blocker metoprolol administered before reperfusion can reduce myocardial infarct size in mice, pigs, and humans by eliminating exacerbated inflammation [102]. PKC also plays an important role during myocardial I/R in redox regulation (redox signaling and oxidative stress), cell death (apoptosis and necrosis), Ca^2+^ overload, and mitochondrial dysfunction [23]. In the present study, the mechanism of safflower treatment in MIR injury was an inflammatory response mediated by the NFκB, HIF-1α, MAPK, TNF, and PI3K/AKT signaling pathways. Safflower markedly inhibited the expression of inflammatory factors, including IL-6, IL-1β, IL-18, MCP-1, and TNF-α.

In the present study, we identified 79 compounds in safflower using UPLC-QTOF-MS/MS. UPLC-QTOF-MS/MS is a powerful tool for the qualitative characterization of chemical components in herbs because it provides high chromatographic and mass resolution, accurate mass measurement, and abundant fragment ion information [20]. Compared with other mass spectrometers, the TOF analyzer has a higher mass resolution, sensitivity, and accuracy; in addition, it can provide accurate ion mass and molecular formulas. MS^E^ technology is a new data acquisition method. It is helpful for the comprehensive analysis of complex samples, and can obtain accurate mass measurements of precursors and product ions at a significant speed [21]. For the first time, we used the UPLC-QTOF-MS/MS strategy to identify safflower phytochemicals.

Safflower is widely used in the clinical treatment of cardio-cerebrovascular diseases. The commonly used safflower preparations include safflower yellow injection, safflower injection, and safflower oil (p.o.). Safflower preparations show excellent protection and safety in treating coronary heart disease, angina pectoris, obesity, and blood pressure [103,104,105,106]. Safflower, as the gentleman medicine of Xuebijing injection, has apparent clinical effects on sepsis [107]. Danhong injection is a medicinal preparation based on *Salviae Miltiorrhizae* and *Flos Carthami* (safflower) and has also been used in the clinical therapy of cardiovascular and cerebrovascular diseases in China for many years [10]. Flavonoids are the main active components of safflower [108]. In the present study, we identified 53 flavonoids out of the 79 compounds in safflower. Flavonoids possess anti-oxidant, anti-microbial, and anti-platelet aggregation effects; they are also recognized as excellent anti-inflammatory agents [109,110]. Flavonoids can inhibit the activation of inflammatory pathways, such as the NFκB, MAPK, and bone morphogenetic protein 2/small mothers against decapentaplegic (BMP2/SMAD) signaling pathways. In addition, they inhibit the expression of pro-inflammatory enzymes, such as activating protein-1, cyclooxygenase-2, lipoxygenase, and inducible nitric oxide, and decrease the expression of various pro-inflammatory cytokines [111,112].

In the present study, we filtered out 11 core anti-MIR injury compounds, including quercetin, luteolin, apigenin, rutin, HSYA, kaempferol, baicalin, eriodictyol, 6-hydroxyapigenin, 6-hydroxykaempferol, and 6-hydroxykaempferol 3-rutinoside-6-glucoside. These compounds belong to the family of flavonoids and flavonoid glycosides. Quercetin is a characteristic flavonoid that has been extensively studied. Quercetin exhibits significant pharmacological effects such as anti-inflammatory, anti-oxidant, anti-viral, and cardioprotective effects [113,114]. Furthermore, quercetin is involved in the NFκB and MAPK signaling pathways and inhibits pro-inflammatory factors. Luteolin is present in vegetables and fruits and is known to be responsible for its anti-inflammatory activity [115,116]. Luteolin inhibits the expression of IL-1β, IL-2, IL-6, IL-8, IL-12, IL-17, TNF-α, and interferon (IFN)-β. Kaempferol also potently inhibits pro-inflammatory proteins such as PKC, NFκB, and MAPK (ERK, p38, and JNK) [117]. Apigenin promotes different anti-inflammatory pathways, such as p38/MAPK and PI3K/AKT [118]. Other flavone compounds also show excellent anti-inflammatory activities [119,120].

HSYA, rutin, and 6-hydroxykaempferol 3-rutinoside-6-glucoside are flavonoid glycosides (quinochalcone C-glycosides) that are characteristic ingredients of safflower [121,122,123]. HSYA, a quality marker (Q-marker) for safflower, is a representative quinochalcone C-glycoside. It exerts anti-oxidant, anti-inflammatory, anti-coagulant, anti-cancer, and cardio-cerebrovascular protective effects. HSYA attenuates the activation of NFκB, MAPK, and Nrf-2/HO-1 signaling pathways [124,125]. Rutin is involved in p53 expression and in the PI3K/AKT signaling and NFκB signaling pathways [126]. To date, 23 quinochalcone C-glycosides have been isolated from safflowers [6,123,125], including safflower yellow B, carthamin, hydroxyethylcarthamin, safflomin A, safflomin B, safflomin C, isoflurane C (isosafflomin C), pre-carthamin, anhydrosafflor yellow B (AHSYB), nitrogen-containing quinochalcone C-glycoside tinctormin and cartormin, saffloquinoside A, saffloquinoside B, saffloquinoside C, methylsafflomin C, methylisosafflomin C, hydroxysafflor yellow B (HSYB), hydroxy red anthocyanin C (HSYC), carthorquinoside A, carthorquinoside B, and isocartormin, all of which possess significant therapeutic potentials [127,128]. However, flavonoids are poorly absorbed, with an extremely low oral bioavailability and a short half-life. Their plasma concentrations in the human body are usually < 1 μmol/L, which presents great challenges for clinical application [129,130,131].

We predicted the mechanism of action of safflower by using TCMIP v2.0. The advantages of the TCMIP are summarized in three aspects. The first is a combination of computational biology and network pharmacology. TCMIP is carried out from the perspective of computer virtualization for the interaction among big data. The second is experimental verification. TCMIP pays more attention to pharmacological evaluation to verify it from a “practical” perspective. Thirdly, the integration of pharmacokinetics and pharmacodynamics. To study the interaction between TCM prescriptions and the body from multiple levels and links and systematically and comprehensively reveal the pharmacodynamic material basis and mechanism of the efficacy of TCM prescriptions, we performed a comprehensive and systematic evaluation by integrating virtual prediction and experimental verification to increase data accuracy [13,132]. To avoid the homogenization of key components screening of “different diseases and different prescriptions”, we addressed the current gaps in the literature of integrated pharmacology, and used UPLC-QTOF-MS/MS to supplement the deficiencies of the database. Integrated pharmacology is a qualitative analysis based on the “component–target–disease” network that ignores the influence of component quantifications. Therefore, we increased the screening of component contents.

## 4. Materials and Methods

### 4.1. Data Preparation

#### 4.1.1. Preparation of Safflower

Safflower (210616z11) was purchased from Beijing Shengshilong Pharmaceutical Co., Ltd., (Beijing, China). The safflower powder was sieved through a 50-mesh sieve and 4.0 g of it was soaked in 50 mL of 25% methanol before ultrasonic extraction for 40 min (power 300 W, frequency 50 kHz). After the supernatant was centrifuged at 3000× *g* rpm for 10 min, it was filtered through a 0.22 μm filter (Pall Corporation, Beijing, China). Then, 2 μL aliquots were injected into the UPLC-QTOF-MS/MS system. Metoprolol (H32025391; AstraZeneca, Switzerland) was used as the positive control.

#### 4.1.2. Animals

All procedures were approved by the Medicine Ethics Review Committee for Animal Experiments at the Institute of Chinese Materia Medica, China Academy of Chinese Medical Sciences. C57/BL6 male mice (specific pathogen-free (SPF) grade, Certification No. 111251220100022578, Ethical No. 2022B039), weighing 20 ± 2 g and eight weeks old, were purchased from Beijing Huafukang Bioscience Co. (Beijing, China). The mice were housed in a controlled environment (24 ± 1 °C temperature, 50 ± 10% relative humidity) with a 12/12 h light/dark cycle and free access to water and standard diet under specific pathogen-free (SPF) conditions.

### 4.2. Pharmacological Evaluation

#### 4.2.1. Induction and Treatment of MIR-Injured Mice

A mouse model of MIR injury was established by LAD ligation as previously described [15]. The mice were anesthetized with 3% isoflurane (Beijing ZS Dichuang Technology Development Co., Ltd., Beijing, China) inhalation using a respiratory anesthesia machine (ZS-MV, Beijing ZS Dichuang Technology Development Co., Ltd., Beijing, China). Afterward, the mice were transferred to 1% isoflurane for maintenance anesthesia. After the pericardium was opened, the heart was exposed between the third and fourth intercostal space on the left, and 2–3 mm of the coronary artery was ligated using a 7–0 silk suture. After 30 min of ligation, reperfusion was performed for 24 h. Mice in the sham group without LAD ligation were also subjected to reperfusion as described above. A total of 126 mice were randomly divided into six groups, including the sham group, control group, safflower low-dose group, medium-dose group, high-dose group (62.5 mg/kg, 125 mg/kg, 250 mg/kg, 4 times/24 h, i.v.), and positive control metoprolol group (12.5 mg/kg/d, i.p.) [11,133]. Among them, 36 mice (six in each group) were used to detect echocardiography, biochemical markers, and qRT-PCR. A total of 54 mice (9 from each group) were assigned to the TTC/Evans blue staining experiment. Western blot and TUNEL experiments (three in each group) included 18 mice in each experiment. Safflower extract and metoprolol were all dissolved in 0.9% NaCl. It was immediately injected into the tail vein injection after ischemia and before reperfusion. Because of rapid elimination within 6 h of safflower [134], it was administered every 6 h. Metoprolol was administered via intraperitoneal injection (12.5 mg/kg/d) [102]. Mice in the sham and control groups were treated with the solvent carrier.

#### 4.2.2. Echocardiography

After 24 h of reperfusion, the mice were transferred to 1% isoflurane for maintenance anesthesia. Cardiac function was evaluated by echocardiography (VisualSonics VeVo 2100 Imaging System). Each group had six mice.

#### 4.2.3. TTC/Evans Blue Staining

Mice were anesthetized with 1% pentobarbital sodium after 24 h reperfusion. 200 μL of 2% Evans blue (Sigma, E2129, Steinheim, Germany) was perfused through thoracic aorta for 30 s. Subsequently, the heart was immediately harvested and frozen in a −80 °C refrigerator for 30 min. Afterward, the heart was sectioned into 2 mm thick slices below the ligation position using heart mold and stained with 1% TTC (Sigma, T8877-100G, USA) at 37 °C for 10 min. Each group had nine mice.

#### 4.2.4. Detection of Biochemical Markers in the Blood

After 24 h of reperfusion, the mice were anesthetized with 1% pentobarbital sodium. Blood was collected from the inferior vena cava and centrifuged at 8000× *g* rpm under 4 °C for 5 min. The plasma in the upper layer was harvested for the detection of SOD (Beyotime, S0101S, Beijing, China) and LDH (Solarbio, BC0685, Beijing, China) activities.

#### 4.2.5. TUNEL Assay

After 24 h of reperfusion, the mice were anesthetized with 1% pentobarbital sodium. The hearts were harvested after perfusing with 5 mL PBS. TUNEL staining was performed using the In Situ Cell Death Detection Kit, Fluorescein (Roche, Beijing, China). The sections (4 µm) were deparaffinized using xylene, rehydrated in 100%, 90%, 80%, and 70% ethanol, and permeabilized in 0.1% Triton-X-100 for 8 min at 37 °C. We mixed 50 μL terminal deoxynucleotidyl transferase (TdT) and 450 μL fluorescein-labeled dUTP solution. A 20 µL volume of staining solution was added per sample. Apoptotic cells were detected after incubation with the mixture for 30 min at 37 °C. The nuclei were labeled with DAPI (ZSGB-BIO, ZLI-9556, Beijing, China).

### 4.3. Data Collection

#### 4.3.1. Component Identification

Chromatography was performed using a Waters UPLC I-Class system (Waters Corp., Milford, MA, USA) equipped with a binary pump, online vacuum degasser, autosampler, and automatic thermostatic column oven coupled with a quadrupole-time-of-flight mass spectrometer. A Waters Xevo G2-S Q-TOF Mass System (Manchester, United Kingdom) equipped with electrospray ionization (ESI). Data were recorded using Masslynx V4.1 (Waters Corporation, Milford, MA, USA). The UNIFI software 1.8 (Waters Corporation, Milford, MA, USA) was used for peak detection and preliminary compound identification. Chromatographic separation was performed on a Waters Acquity UPLC HSS T3 column (100 mm × 2.1 mm, i.d., 1.8 μm) maintained at 30 °C, and a linear gradient of 0.1% formic acid–water (A) and 0.1% formic acid–acetonitrile (B) was used for the elution procedure, as follows: 0–2 min, 5–90% B; 2–10 min, 90–80% B; 10–16 min, 80–60% B; 16–20 min, 60–5% B. The flow rate was set at 0.2 mL/min, and a 2.0 µL aliquot was set as the injection volume.

#### 4.3.2. Mass Spectrometry Conditions

The UPLC-QTOF-MS/MS data were collected in full-scan auto mode in positive and negative ion modes. The optimal parameters for the best response for most of the compounds were set as follows: ESI^+^ capillary voltage, 0.5 KV; ESI^-^ capillary voltage, 2.5 KV; sampling cone, 40 V; source temperature, 100 °C; desolvation temperature, 450 °C; gas temperature of atmospheric gas, 450 °C; cone gas flow, 50 L/h; desolvation gas flow, 900 L/h; mass range, 50–1, 500 *m*/*z*. The collision energies were 40–60 V for ESI^+^ and 60–80 V for ESI^−^.

#### 4.3.3. Data Processing

Waters UNIFI 1.8 data processing software was used to process the quasi-molecular ion peaks collected using the UPLC-QTOF-MS/MS system. The MS^E^ data collected in a continuum mode were processed and matched to a customized library based on the Encyclopedia of Traditional Chinese Medicine (ETCM, http://www.nrc.ac.cn:9090/ETCM/, accessed on 2 January 2022) using the Waters UNIFI system with an error of 1 × 10^−5^ ppm. The analysis process included data acquisition, data mining, library searching, and report generation.

### 4.4. Mechanism Prediction

#### 4.4.1. Prediction of the Targets of Safflower

The chemical structural formula (sdf.) of identified compounds in safflower were collected from PubChem (https://pubchem.ncbi.nlm.nih.gov/, accessed on 20 January 2020), which were transformed into *mol.* formula by OpenBabel GUI 2.4.1 (last update on 21 September 2016, version 2.4.1, http://openbabel.org/wiki/MainPage, accessed on 20 January 2022). And then uploaded them to TCMIP v2.0 (http://www.tcmip.cn, accessed on 20 January 2020) to predict putative targets with a Tanimoto score of 0.7.

#### 4.4.2. Collection of MIR-Related Targets

MIR-related targets were collected from three well-known databases with the keywords of “Myocardial ischemia–reperfusion injury,” “Myocardial ischemia injury,” “Acute myocardial ischemia injury,” and “Myocardial infarct injury”: the DisGeNET database (http://www.disgenet.org, accessed on 20 January 2022), GeneCards: The Human Gene Database (https://www.genecards.org/, accessed on 20 January 2022), and TCMIP v2.0. Scores > 0.1 for DisGeNET, scores > 10 for GeneCards, and no threshold for TCMIP v2.0. We combined all the genes, removed duplicates, and obtained MIR-related targets. Detailed information is provided in Appendix A.

#### 4.4.3. Functional Analysis and Network Construction

To better demonstrate the common targets of safflower and MIR injury, the predictive targets of safflower and MIR-related genes were uploaded to the website of the Venn diagram (Appendix A) (http://bioinformatics.psb.ugent.be/webtools/Venn/, accessed on 22 January 2022). Overlapping genes from the two groups were used for further analysis. The potential biological function of safflower was analyzed by KEGG and GO enrichment using the DAVID v6.8 database (Appendix A) (https://david.ncifcrf.gov/, accessed on 30 January 2022). The top 30 BP terms and 34 KEGG terms were shown in Figure 4. To clearly explain the complex relationships between components of safflower, known MIR-related genes, and predicted signaling pathways, network visualization was performed using the CytoScape platform (version 3.9.0, https://cytoscape.org/, accessed on 30 January 2022). Components, genes, and pathways were all present as independent nodes (Appendix A). CytoScape 3.9.0 calculated three topological parameters for each of these nodes, including “degree”, “betweenness”, and “closeness”.

#### 4.4.4. Hub Target and Key Component Screening

(1)Hub target screening:

Hub target screening was performed using network topological analysis. The hub genes were defined as having higher degree values than the median. For genes, the median degree value is 6.

(2)Key component screening:

To avoid homogenization, key component screening was subjected to the three screening criteria described below, and the components that met two of the three screening criteria are regarded as core compounds. The details were listed as follows:①Network topological analysis: The degree value of the network was calculated to screen for core compounds. The degree values of the core compounds were higher than the median. For components, the median degree value is 8;②Literature mining: A systematic search was performed on PubMed using the following sets of keywords: the “name” of the ingredients, “myocardial ischemia–reperfusion,” and “cardiac ischemia reperfusion.” Studies included in this search were those published from 1967 to November 2022. The literature mainly focuses on animal studies, and none of the ingredients has been found to be used clinically for the treatment of MIR injury. Articles related to the combinations of ingredients were also included. In the selected studies, the following data were meticulously reviewed and extracted: “infarct size detection,” “cardiac function detection” or “serum parameters.” The median of the number of studies was calculated to filter the key compounds;③Quantitative analysis: The response values of the compounds were ranked according to UPLC-QTOF-MS/MS to screen for components with response values above the median.

#### 4.4.5. Molecular Docking Simulation

The molecular docking and virtual screening program were carried out to investigate the direct binding efficiencies of hub targets and key components. PRKCA (PDB ID: 3iw4), AKT1 (PDB ID: 3qkm), and PI3K (PDB ID: 1e7u) were collected from the Protein Data Bank (https://www.rcsb.org/, accessed on 20 January 2023). The ligands of 6-hydroxyapigenin, 6-hydroxykaempferol-3-rutinoside-6-glucoside, 6-hydroxykaempferol, apigenin, baicalin, eriodictyol, HSYA, kaempferol, luteolin, quercetin, and rutin were downloaded from the PubChem database (31) in sdf. format, which were converted to pdb. format using OpenBabel GUI 2.4.1. AutoDock Tools 1.5.6 (https://ccsb.scripps.edu/mgltools/, accessed on 22 January 2023) was used for dehydration, hydrogenation, and charging. Docking calculations were performed using AutoDock Vina 1.1.2 (The Scripps Research Institute) and AutoDock 4.2.6 (The Scripps Research Institute). The visualization and analysis of the results were used by PyMOL 2.5 (https://pymol.org/2/, accessed on 22 January 2023).

### 4.5. Mechanism Verification

#### 4.5.1. Quantitative Real-Time Reverse Transcription-Polymerase Chain Reaction (qRT-PCR)

qRT-PCR was performed using the SYBR^®^ Select Master Mix (Toyobo, Tokyo, Japan). A total RNA was extracted using TRNzol Universal Reagent (Tiangen, DP430, Beijing, China), and 300 ng RNA was reverse-transcribed into cDNA using ReverTra Ace qPCR RT Master Mix (TOYOBO, FSQ-301, Osaka, Japan). Single-stranded cDNA was amplified by PCR with primers for NFκBia, NFκB1, IL6, IL1β, MCP-1, IL-18, TNF-α, and β-actin using Taq pro Universal SYBR qPCR Master Mix (Vazyme, Q712-02, Nanjing, China); the primer sequences are shown in Table 3. Primers were synthesized by Tsingke Biotechnology Co., Ltd. (Beijing, China).

#### 4.5.2. Western Blot

After 24 h of reperfusion, the hearts of mice were collected for Western blot analysis. The total protein content of the supernatant was quantified using Bicinchoninic acid (BCA) protein assay kit (Beyotime, P0010S, Nanjin, China). Protein samples were boiled at 100 °C in 5 × SDS loading buffer (Beyotime, P0015) for 10 min. The proteins of 30 μg were run in 8%, 10%, and 12% gradient SDS-PAGE at 80 V for 30 min, then converted to 120 V for 60 min; afterward, they were transferred onto 0.45 μm PVDF membranes (Sigma, HVLP02500) under 200 mA for 1 h, blocked in 5% milk (Sigma, 20-200) for 1.5 h, incubated in primary antibodies overnight at 4 °C, then incubated with secondary biotinylated antibodies for 2 h at room temperature. Proteins were detected with ECL (32132, Thermo Fisher, Carlsbad, CA, USA). The antibodies were listed as follows: HIF1α (BIOSS, bs-20399R), VEGFA (BIOSS, bs-0279R), PI3K (Proteintech, 60225-1-Ig, Rosemont, IL, USA), phospho-PI3K (Cell Signaling Technology, 4228, Danvers, MA, USA), AKT (BIOSS, bs-0115R), phospho-AKT(Cell Signaling Technology, 4060), PKCε (Abclonal, A4998, Woburn, MA, USA), ERK1/2 (Abclonal, A10613), phospho-ERK1/2 (Abclonal, AP0472), p-NFkB-p65 (Cell Signaling Technology, 3033), NFkB-p65 (Abclonal, A18210), BAX (Proteintech, 50599-2-Ig, Rosemont, IL, USA), BCL2 (Proteintech, 68103-1-Ig), β-tubulin (Proteintech, 10094-1-AP), anti-rabbit IgG, HRP-linked antibody (Cell Signaling Technology, 7074), and anti-mouse IgG, HRP-linked antibody (Cell Signaling Technology, 7076).

### 4.6. Statistical Analysis

Data were analyzed by one-way analysis of variance (ANOVA) using GraphPad Prism 8.0.1 software. Statistical significance was set at *p* < 0.05. Data are shown as mean ± SEM.

## 5. Conclusions

We employed an integrative pharmacological strategy to explore the mechanism of action of safflower in improving MIR injury in mice. We characterized 79 chemical components of safflower. Among them, 56 chemical compounds, including 11 key ingredients, may ameliorate MIR injury partially by interacting with 31 hub candidate targets, mainly through an “inflammation-immune” system. Further studies are needed to conduct more systematic efficacy evaluations and mechanistic explorations.

## Figures and Tables

**Figure 1 ijms-24-05313-f001:**
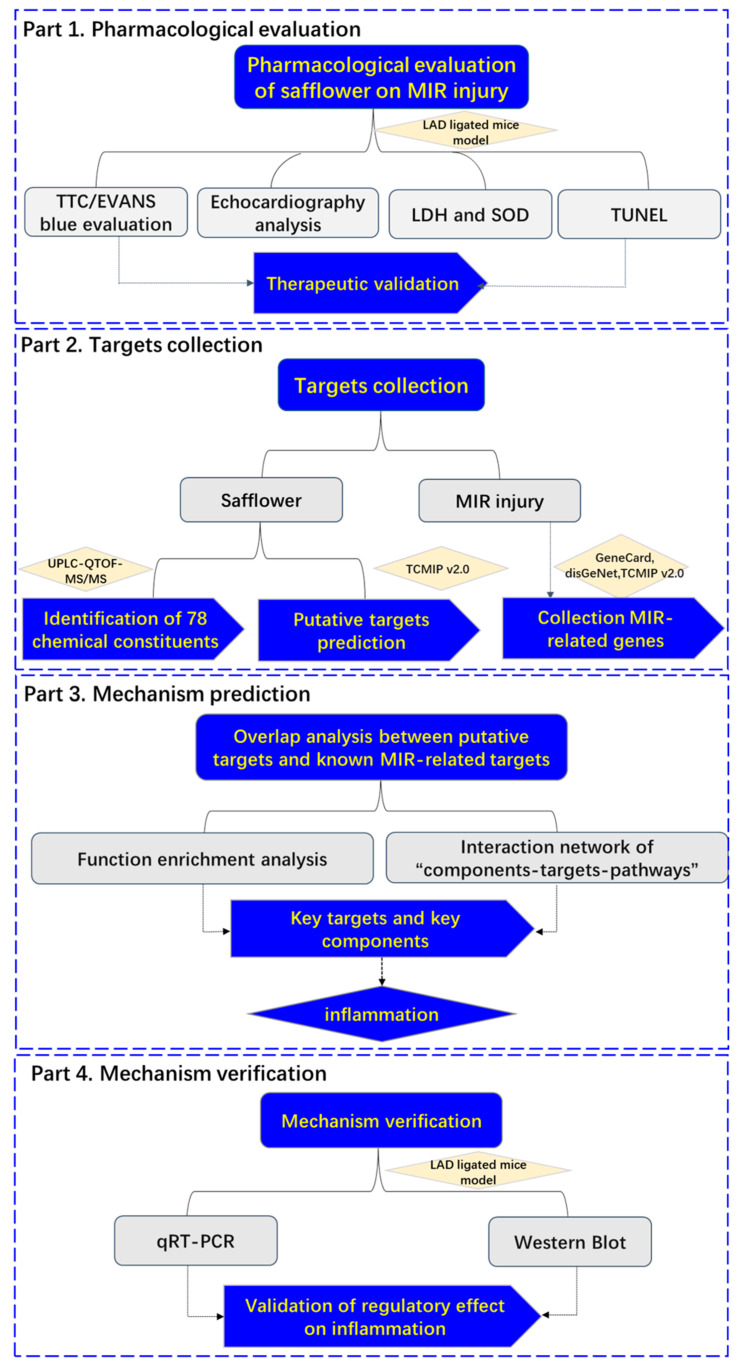
A schematic diagram of the systematic strategies for uncovering the pharmacological mechanisms of safflower action on MIR injury.

**Figure 2 ijms-24-05313-f002:**
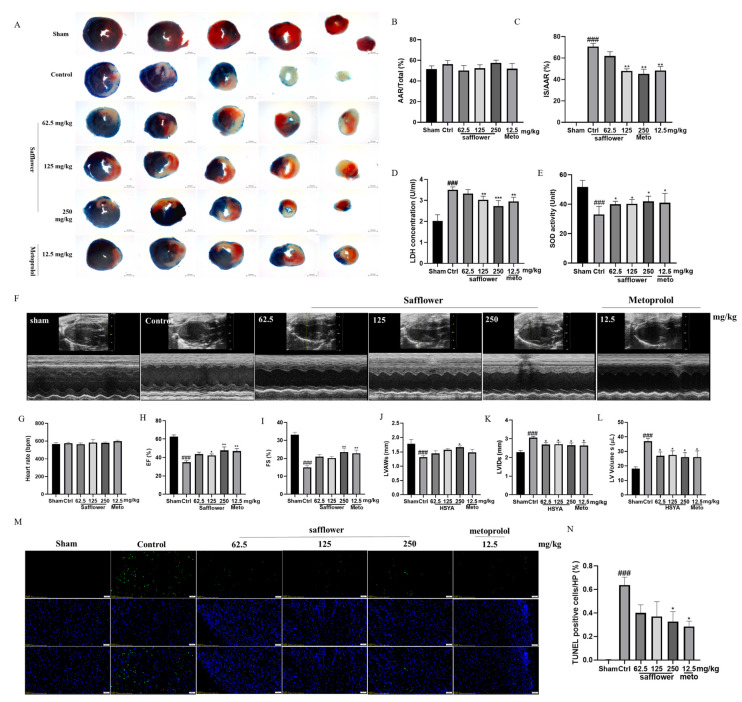
Pharmacodynamic studies of safflower effect on MI/R injury. Representative photographs of TTC/Evans blue perfused hearts were shown for sham, control, safflower treatment (62.5, 125, 250 mg/kg, 4 times/24 h) and metoprolol groups (12.5 mg/kg/24 h) (**A**, scale bar is 200 pixels). Infarct size (IS) (**C**) and area at risk (AAR) (**B**) were quantified after MIR injury for 24 h (*n* = 9). Representative echocardiograph depicted for sham, control, safflower treatment, and metoprolol treatment (**F**) (*n* = 6). Rate (**G**), EF (**H**), FS (**I**), LVAWs (**J**), LVIDs (**K**), and LV Volume s (**L**) values were quantified. Average serum LDH (**D**) (*n* = 6) and SOD (**E**) (*n* = 6) activities in MIR injury mice assayed on reperfusion for 24 h. Representative photographs of TUNEL staining (**M**, scale bar is 50 μm) and the statistical diagram of quantification (**N**) were shown (*n* = 3). One-way ANOVA with Tukey’s multiple comparison test. Values are means ± standard error of mean (SEM), *^###^ p* < 0.001 vs. sham, respectively; * *p* < 0.05, ** *p* < 0.01, **** p* < 0.001 vs. control, respectively; one-way ANOVA.

**Figure 3 ijms-24-05313-f003:**
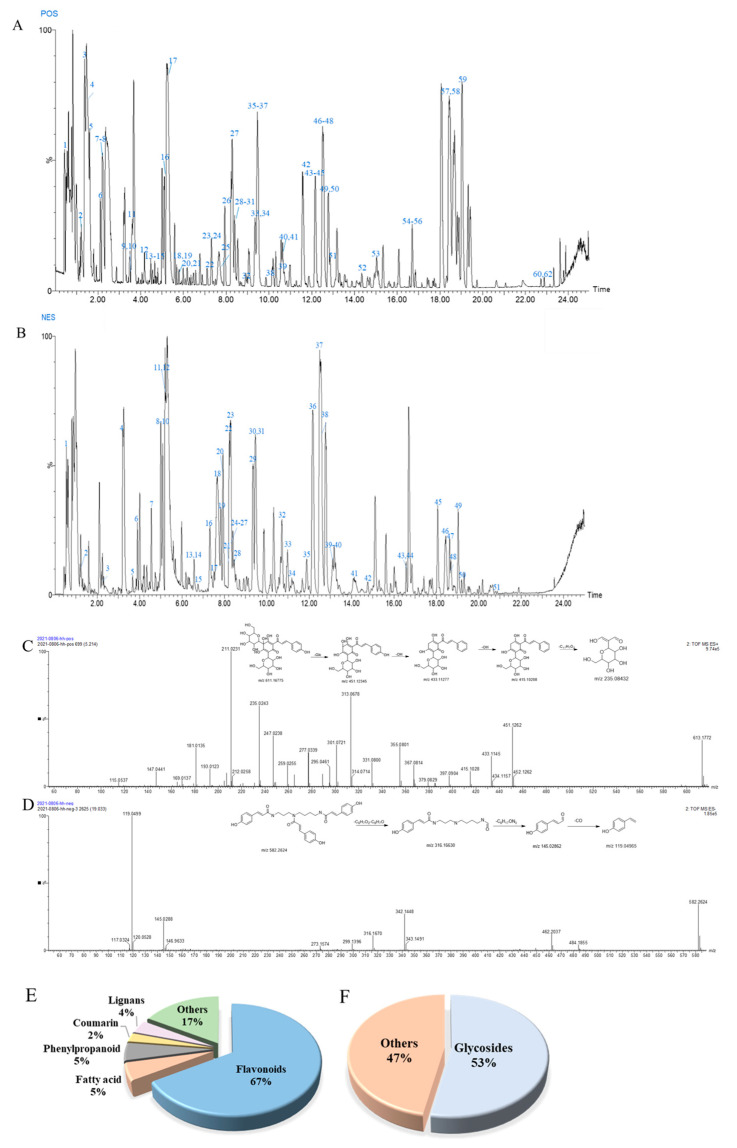
HPLC/QTOF/MS/MS chromatograms of water extracts of safflower in ESI^+^ (**A**) and ESI^-^ (**B**) ion mode. Spectrum information of HSYA (**C**) and N1, N5,N10-(Z)-tri-p-cou-maroylspermidine (**D**) automatically provided by UNIFITM. Structural classification is shown (**E**,**F**).

**Figure 4 ijms-24-05313-f004:**
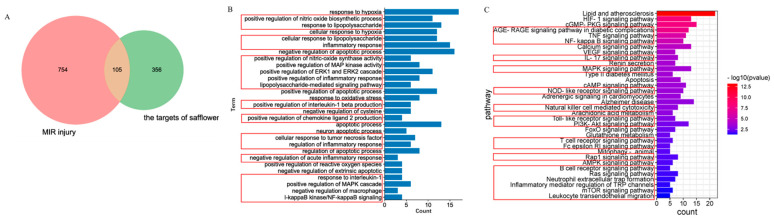
Venn diagram of drug targets and disease genes (**A**) are shown. GO (**B**) and KEGG (**C**) enrichment analysis of safflower in the treatment of MIR injury are shown. The terms labeled by using red boxes were associated with inflammation.

**Figure 5 ijms-24-05313-f005:**
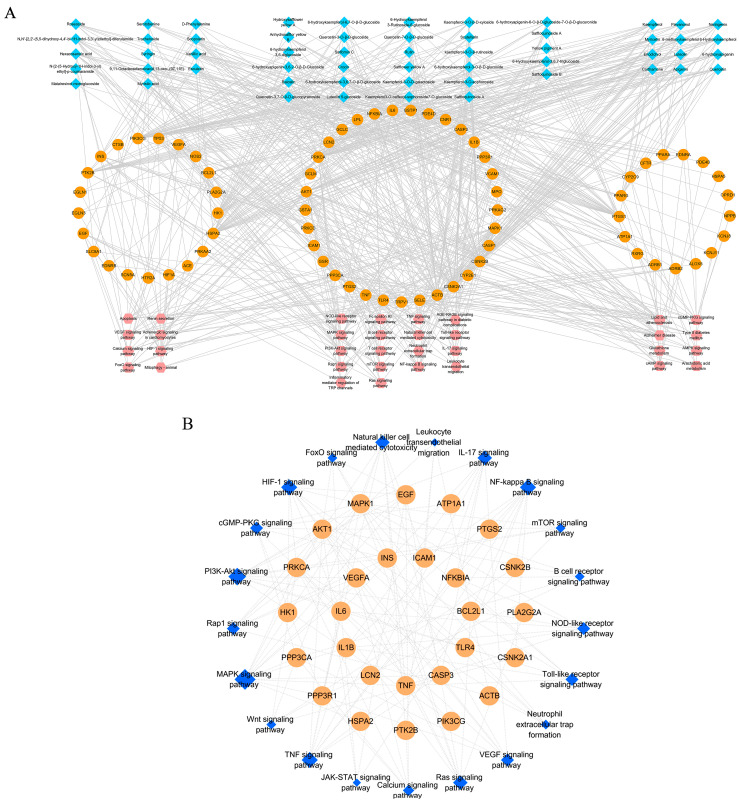
The interactive network of “components-targets-pathways-symptoms” (**A**) was shown. The network of hub targets and key pathways (**B**) were also shown. In (**A**), yellow nodes refer to the safflower, blue nodes refer to putative targets of safflower, orange nodes refer to therapeutic targets of MIR, and pink nodes refer to KEGG pathway. In (**B**), orange nodes refer to hub targets filtered from (**A**), and the blue one refers to the key pathway.

**Figure 6 ijms-24-05313-f006:**
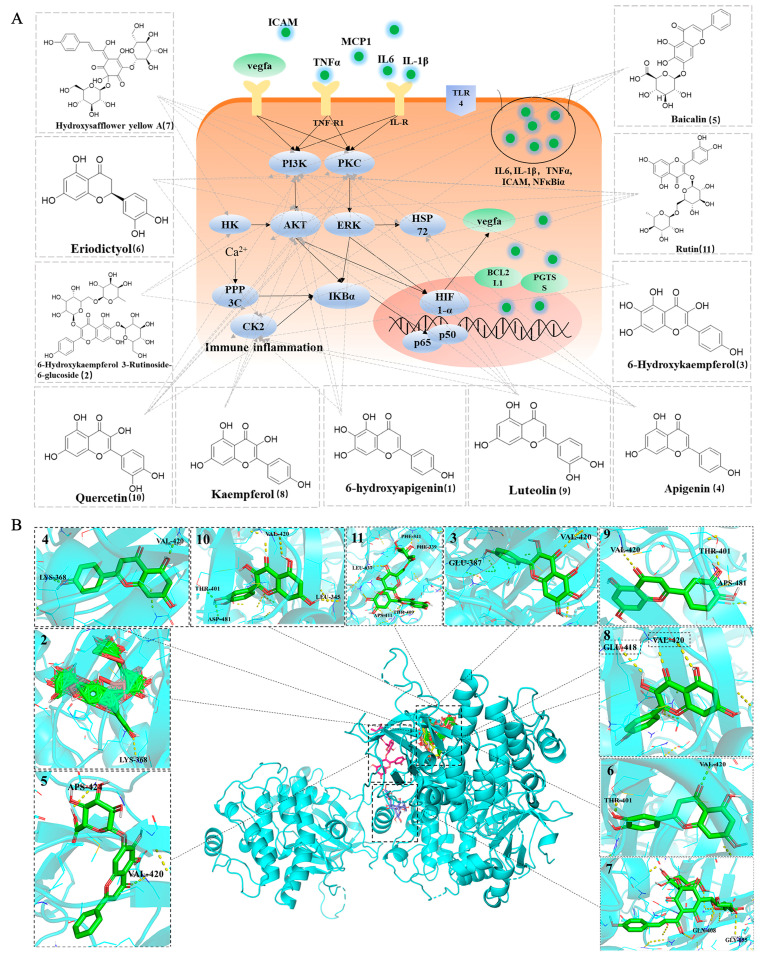
A diagram of mechanism of safflower in the treatment of MIR injury with the hub targets, key components, and crucial signaling pathways (**A**). Molecular docking simulation of the binding patterns of PRKCA (**B**), PIK3CG (**C**), and AKT1 (**D**) with the key components, including 6-hydroxyapigenin, 6-hydroxykaempferol-3-rutinoside-6-glucoside, 6-hydroxykaempferol, apigenin, baicalin, eriodictyol, HSYA, kaempferol, luteolin, quercetin, and rutin, respectively. Caption: 1. 6-hydroxyapigenin; 2. 6-Hydroxykaempferol-3-rutinoside-6-glucoside; 3. 6-Hydroxykaempferol; 4. Apigenin; 5. Baicalin; 6. Eriodictyol; 7. HSYA; 8. Kaempferol; 9. Luteolin; 10. Quercetin; 11. Rutin.

**Figure 7 ijms-24-05313-f007:**
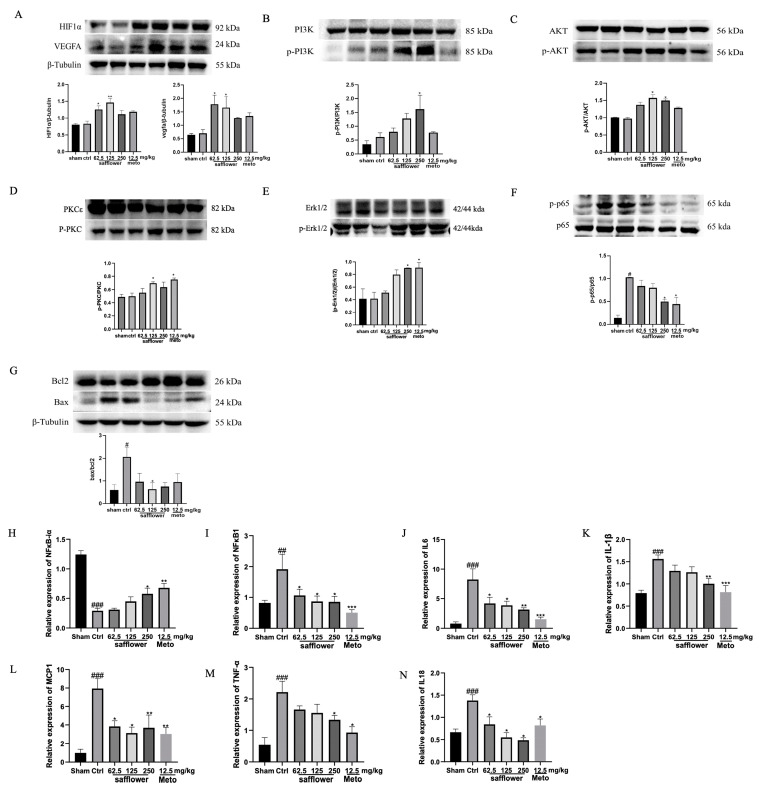
The protein expression of HIF1α (**A**), VEGFA (**A**), p-PI3K/PI3K (**B**), p-AKT/AKT (**C**), p-PKC/PKC (**D**), p-ERK1/2/ERK1/2 (**E**), p-NFkB-p65/NFkB-p65 (**F**), and BAX/BCL2 (**G**) in the hearts of MIR-injured mice using Western blotting analysis (*n* = 3 per group). The mRNA expressions of inflammation-related factors in safflower treatment of MIR injury are shown (**H**–**N**). Values are means ± SEM, ^#^
*p* < 0.05, ^##^
*p* < 0.01, *^###^ p* < 0.001 vs. sham, respectively; * *p* < 0.05, ** *p* < 0.01, **** p* < 0.001 vs. control, respectively; one-way ANOVA.

**Table 1 ijms-24-05313-t001:** The key targets screened from the network analysis.

Targets	Degree	Closeness	Betweenness
PRKCA	46	0.46939	0.13402
AKT1	38	0.45609	0.11656
PIK3CG	36	0.39558	0.04650
CSNK2A1	36	0.38609	0.03410
CSNK2B	36	0.38609	0.03410
HSPA2	35	0.38983	0.03284
PTGS1	27	0.38609	0.08554
MAPK1	26	0.42037	0.05791
HK1	23	0.34773	0.03794
TNF	19	0.36508	0.02730
PTK2B	19	0.34183	0.01179
INS	14	0.36842	0.01911
PPARG	13	0.35698	0.01304
IL1β	13	0.34183	0.01452
IL6	13	0.35385	0.01586
PPP3R1	13	0.37182	0.01717
PPP3CA	13	0.37182	0.01717
ATP1A1	12	0.33472	0.05071
BCL2L1	12	0.33612	0.00845
NFκBIA	11	0.35385	0.00975
PPARA	11	0.34328	0.00984
PTGS2	10	0.34773	0.01090
CASP3	10	0.33612	0.00581
TLR4	10	0.35698	0.01096
VEGFA	9	0.35076	0.00812
LCN2	8	0.32525	0.00361
PLA2G2A	8	0.34475	0.01259
EGF	8	0.33895	0.01847
CYP2C9	8	0.33612	0.00471
TRPV1	7	0.32394	0.01434
ICAM1	7	0.32924	0.00425

**Table 2 ijms-24-05313-t002:** The key components filtered from the network.

Component	Targets	Degree	Response	Literature Mining
Quercetin (+40)	PRKCA, AKT1, ACTB, CSNK2A1, CSNK2B, PIK3CG, HSPA2, PTK2B, PPARG, BCL2L1, PPARA, LCN2	12	233,661	17 [24,25,26,27,28,29,30,31,32,33,34,35,36,37,38,39,40]
Luteolin (-44, +55)	PRKCA, AKT1, ACTB, CSNK2A1, CSNK2B, PIK3CG, HSPA2, PTK2B, PPARG, BCL2L1, PPARA, LCN2	12	90,094	23 [41,42,43,44,45,46,47,48,49,50,51,52,53,54,55,56,57,58,59,60,61,62,63]
Apigenin (-47, +58)	PRKCA, AKT1, ACTB, CSNK2A1, CSNK2B, PIK3CG, HSPA2, PTK2B	11	175,455	6 [64,65,66,67,68,69]
Rutin (+33)	PRKCA, ACTB, CSNK2A1, CSNK2B, PIK3CG, HSPA2, CYP2C9	7	732,202	4 [25,70,71,72]
HSYA (-11, +17)	PTGS1, HK1	2	4,210,682	8 [73,74,75,76,77,78,79,80]
Kaempferol (-42, +44)	PRKCA, AKT1, ACTB, CSNK2A1, CSNK2B, PIK3CG, HSPA2, PTK2B, PPARG, PPARA	12	876,007	5 [81,82,83,84,85]
Baicalin (-40, +51)	PRKCA, AKT1, ACTB, CSNK2A1, CSNK2B, PIK3CG, HSPA2, PTK2B,	8	73,419	9 [86,87,88,89,90,91,92,93,94]
Eriodictyol (+23)	PRKCA, AKT1, ACTB, CSNK2A1, CSNK2B, PIK3CG, HSPA2, PTK2B, CYP2C9	9	306,217	2 [95,96]
6-hydroxyapigenin (-48, +46)	PRKCA, AKT1, ACTB, CSNK2A1, CSNK2B, PIK3CG, HSPA2, PTK2B, PPARG, BCL2L1, PPARA, LCN2	12	338,578	0
6-Hydroxykaempferol (-32, +35)	PRKCA, AKT1, ACTB, CSNK2A1, CSNK2B, PIK3CG, HSPA2, PTK2B, PPARG, BCL2L1, PPARA, LCN2	12	1,375,662	0
6-Hydroxykaempferol 3-Rutinoside-6-glucoside	ACTB, CSNK2A1, CSNK2B, PIK3CG, HSPA2, PTGS1, HK1	9	1,632,678	0

**Table 3 ijms-24-05313-t003:** Primers used in this study.

Gene	Forward Primer	Reward Primer
NFκBia	CAAATGGTGAAGGAGCTGCG	CCAAGTGCAGGAACGAGTCT
NFκB1	AGCAACCAAAACAGAGGGGA	TGCAAATTTTGACCTGTGGGT
IL6	ACAACCACGGCCTTCCCTACTT	CACGATTTCCCAGAGAACATGTG
IL1β	TGAAGTTGACGGACCCCAAA	TGATGTGCTGCTGTGAGATT
MCP-1	GGCTCAGCCAGATGCAGTTAAC	GCCTACTCATTGGGATCATCTTG
IL-18	CAGGCCTGACATCTTCTGCAA	TCTGACATGGCAGCCATTGT
TNF-α	AAGCCTGTAGCCCACGTCGTA	GGCACCACTAGTTGGTTGTCTTTG
β-actin	CCTGAGCGCAAGTACTCTGTGT	GCTGATCCACATCTGCTGGAA

## Data Availability

The data presented in this study are available in Appendix A.

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
