# Peer review of "Mechanism Repositioning Based on Integrative Pharmacology: Anti-Inflammatory Effect of Safflower in Myocardial Ischemia–Reperfusion Injury"

_ijms, 2023, doi:10.3390/ijms24065313_

Round 1
Reviewer 1 Report (New Reviewer)
The paper, “Mechanism Repositioning Based on Network Pharmacology: Anti-Inflammatory Effect of Safflower on Myocardial Ischemia Reperfusion Injury”, adopted the integrative pharmacology strategy to reveal the mechanisms of safflower treatment in an in vivo model of myocardial ischemia-reperfusion (MIR) injury. The study had combined in silico, experimental work and liquid chromatography/ mass spectrometry, and tried to solve the homogenization phenomenon by literature mining. The manuscript is interesting. However, the revision remains to be answered according to the following suggestions before publication.
Major concerns
Point 1: Network pharmacology is a breakthrough point for traditional medicine research in the era of biomedical big data and artificial intelligence. Network Pharmacology Evaluation Method Guidance plays the guiding role to integrative pharmacology. Whether the study is consistent with this Guidance needs to be clarified. Moreover, authors may check the studies according to these criteria and make the corresponding changes and explanations.
Point 2: The study should increase the discussion on the clinical work of safflower in the treatment of cardiovascular disease.
Point 3: The integrative pharmacology strategy had used in this study, and the differences between it and network pharmacology should be highlighted.
Point 4: The study paid more attention on inflammation, whether to carry out relevant researches in other aspects.
General comments
Point 5: SOD and LDH should be defined in the abstract.
Point 6: Abbreviations should be consistent in the whole paper, such as BCL2 or Bcl2, 1967 (line 551) or 1,967.
Point 7: The part of methodology needs to be supplemented with details.
Point 8: Language revision is needed in somewhere.
Author Response
Response to Reviewer 1 Comments
Major concerns
Point 1: Network pharmacology is a breakthrough point for traditional medicine research in the era of biomedical big data and artificial intelligence. Network Pharmacology Evaluation Method Guidance plays the guiding role to integrative pharmacology. Whether the study is consistent with this Guidance needs to be clarified. Moreover, authors may check the studies according to these criteria and make the corresponding changes and explanations.
Response 1: Thank you so much for your suggestions. We clarified that the study followed the Network Pharmacology Evaluation Method Guidance. We have modified the manuscript as follows [line 322-326]: “According to Network Pharmacology Evaluation Method Guidance, the network pharmacology evaluation is conducted from reliability, standardization, and rationality [1]. In our study, UPLC-QTOF-MS/MS, literature mining and experimental verification ensured the reliability, and TCMIP v2.0 platform supplemented the standardization and rationality”.
Point 2: The study should increase the discussion on the clinical work of safflower in the treatment of cardiovascular disease.
Response 2: Thank you so much for your comments. We have increased the clinical work of safflower in the treatment of cardiovascular disease in the discussion as follows: “Safflower is widely used in clinical treatment of cardio-cerebrovascular diseases. The commonly used safflower preparations include safflower yellow injection, safflower injection, and safflower oil (p.o.). Safflower shows excellent protection and safety in treating coronary heart disease, angina pectoris, obesity, and blood pressure [2-5]”.
Point 3: The integrative pharmacology strategy had used in this study, and the differences between it and network pharmacology should be highlighted.
Response 3: The integrative pharmacology and network pharmacology both based on artificial intelligence and big data, which all integrate massive clinical and experimental data and combining scientific verification to reveal the regulation mechanisms. The integrative pharmacology strategy based on TCMIP v2.0 and ETCM platform, which established qualitative and quantitative pharmacokinetics-pharmacodynamics (PK-PD) correlations through the integration of knowledge from multiple disciplines and techniques and from different PK-PD processes in vivo. And the integrative pharmacology also applied new computational models [6].
Point 4: The study paid more attention on inflammation, whether to carry out relevant researches in other aspects.
Response 4: Thank you so much for your suggestions. We also conducted the experiments on oxidative stress and microvascular obstruction of safflower on MIR-injured mice. The results showed that safflower did not significantly improve the microcirculation after MIR injury (Figure 1, the method is described previously [7]). As for oxidative stress, we found HSYA, the representative component of safflower, has excellent antioxidant effect, which will be further in-depth study.
Figure 1. Safflower could not promote microvascular perfusion following myocardial ischemia reperfusion
Minor comments:
Point 5: SOD and LDH should be defined in the abstract.
Response 5: Thank you so much for your valuable suggestions. We have supplemented the definition of “SOD” and “LDH” in the abstract.
Point 6: Abbreviations should be consistent in the whole paper, such as BCL2 or Bcl2, 1967 (line 551) or 1,967.
Response 6: Thank you so much for your great efforts on our manuscript. We have revised the abbreviations in the whole paper. The “Bcl2” has been revised as BCL2, and the “1967” has been revised as “1,967”. Other changes were also marked in yellow in the manuscript.
Point 7: The part of methodology needs to be supplemented with details.
Response 7: Thank you so much for your positive comments and helpful suggestions on our manuscript. We have revised the section of “5. Materials and Methods”, and supplemented the details of the experimental manipulation: 1. We provided the Item No. in this part (line 470 and 473); 2. We also supplemented the platform and website information (5.4.5).
Point 8: Language revision is needed in somewhere.
Response 8: Thank you so much for your suggestions. We have carefully revised the whole manuscript and hoped the language level has been subs tactically improved.
Reference
[1] Li, S.; Chen, Y.; Ding, Q.; et al. Network Pharmacology Evaluation Method Guidance-Draft. World Journal of Gastroenterology. 2021, 202, 146-154.
[2] Zhang, Q.; Peng, J.; Zhang, X. A clinical study of safflower yellow injection in treating coronary heart disease angina pectoris with Xin-blood stagnation syndrome. Chin J Integr Med. 2005, 11, 222-225.
[3] Lu, Q.; Xu, J.; Li, Q.; Wu, W.; Wu, Y.; Xie, J.; et al. Therapeutic efficacy and safety of safflower injection in the treatment of acute coronary syndrome. Evid Based Complement Alternat Med. 2021, 16, 6617772 (1-10).
[4] Ruyvaran, M.; Zamani, A.; Mohamadian, A.; Zarshenas, M. Eftekhari, M.; Pourahmad, S.; et al. Safflower (Carthamus tinctorius L.) oil could improve abdominal obesity, blood pressure, and insulin resistance in patients with metabolic syndrome: A randomized, double-blind, placebo-controlled clinical trial. J Ethnopharmacol. 2022, 10, 114590 (1-44).
[5] Chen, Y.; Li, M.; Wen, J.; Pan, X.; Deng, Z.; Chen, J.; et al. Pharmacological activities of safflower yellow and its clinical applications. Evid Based Complement Alternat Med. 2022, 27, 2108557 (1-24).
[6] Xu, H.; Zhang, Y.; Wang, P.; Zhang, J.; Chen, H.; Zhang, L.; et al. A comprehensive review of integrative pharmacology-based investigation: A paradigm shift in traditional Chinese medicine. Acta Pharm Sin B. 2021, 11, 1379-1399.
[7] Zhu, L.; Xu, C.; Huo, X.; Hao, H.; Wan, Q.; Chen, H.; et al. The cyclooxygenase-1/mPGES-1/endothelial prostaglandin EP4 receptor pathway constrains myocardial ischemia-reperfusion injury. Nat Commun. 2019, 10, 1888.
Please see the attachment.

Reviewer 2 Report (New Reviewer)
1. Line 303-304 mention mRNA levels of PI3K and HIF1alpha, however, there is no result about that.
2. MAPK signaling pathway is one of the most significant regulated pathway, why only ERK1/2 activation was evaluated, not p38 and JNK?
Author Response
Response to Reviewer 2 Comments
Point 1: Line 303-304 mention mRNA levels of PI3K and HIF1alpha, however, there is no result about that.
Response 1: Thank you so much for your valuable suggestions. We have supplemented the expression of mRNA levels of PI3K and HIF1α in the supplementary file 13. As shown in Figure 1, there were no significant differences in the mRNA levels of PI3K or HIF1α.
Figure 1. The mRNA levels of PI3K and HIF1α
Values are means ± SEM, #P < 0.05, ##P < 0.01, ###P < 0.001 vs. sham, respectively; *P < 0.05, **P < 0.01, ***P < 0.001 vs. Control, respectively; One-way ANOVA.
Point 2: MAPK signaling pathway is one of the most significant regulated pathway, why only ERK1/2 activation was evaluated, not p38 and JNK?
Response 2: Thank you so much for your comments. In our study, the integrative pharmacology strategy was employed to predict the mechanism of action of safflower in treatment on myocardial ischemia reperfusion injury. ERK2 (MAPK1) was filtered from the network analysis as the key target, but not p38 and JNK (Table 1 in the manuscript). In addition, PKC (PRKCA), as the core target, was also screeded in our study. Several studies showed that that PKC overexpression increased ERK1/2 activation without altering other MAP-kinases such as p38 MAPK or JNK [1-4]. Because of the 84% identical in sequence, ERK1 and ERK2 are usually detected together [5]. So, we only evaluated the expression of ERK1/2 in this study.
Reference
[1] Braz, J.; Bueno, O.; De Windt, L.; Molkentin, J. PKC alpha regulates the hypertrophic growth of cardiomyocytes through extracellular signal-regulated kinase1/2 (ERK1/2). J Cell Biol. 2002, 156, 905-919.
[2] Zhang, Y.; Guo, Z.; Li, M.; Fong, P.; Zhang, J.; Zhang, C.; et al. Gabapentin effects on PKC-ERK1/2 signaling in the spinal cord of rats with formalin-induced visceral inflammatory pain. PLoS One. 2015, 29, e0141142.
[3] Wang, H.; Ferraris, J.; Klein, J.; Sands, J.; Burg, M.; Zhou, X. PKC-α contributes to high NaCl-induced activation of NFAT5 (TonEBP/OREBP) through MAPK ERK1/2. Am J Physiol Renal Physiol. 2015, 308, F140-148.
[4] Lee, Y.; Soh, J.; Jeoung, D.; Cho, C.; Jhon, G.; Lee, S.; et al. PKC epsilon -mediated ERK1/2 activation involved in radiation-induced cell death in NIH3T3 cells. Biochim Biophys Acta. 2003, 1593, 219-229.
[5] Roskoski, R. ERK1/2 MAP kinases: structure, function, and regulation. Pharmacol Res. 2012, 66, 105-143.
Please see the attachment.

Round 2
Reviewer 1 Report (New Reviewer)
1. For the sentence in the text: "This approach will be the next promising paradigm shift, from the "one target, one component" to the "network targets, multi-component." Please cited the original reference: Network target for screening synergistic drug combinations with application to traditional Chinese medicine. BMC Systems Biology 2011
2. Please compare the results with the network pharmacology studies regarding herbal formulae containing Safflower, for example, Xuebijing injection (PMID: 33799226).
Author Response
Response to Reviewer 1 Comments
Point 1: For the sentence in the text: "This approach will be the next promising paradigm shift, from the "one target, one component" to the "network targets, multi-component." Please cited the original reference: Network target for screening synergistic drug combinations with application to traditional Chinese medicine. BMC Systems Biology 2011.
Response 1: Thank you so much for your comments. We have supplemented the original reference in the manuscript as follows: This approach will be the next promising paradigm shift, from the "one target, one component" to the "network targets, multi-component" [1].
Point 2: Please compare the results with the network pharmacology studies regarding herbal formulae containing Safflower, for example, Xuebijing injection (PMID: 33799226).
Response 2: Thank you so much for your suggestion. Safflower is widely used in clinical treatment of cardio-cerebrovascular diseases. The commonly used safflower preparations include safflower yellow injection, safflower injection, and safflower oil (p.o.). Safflower preparations show excellent protection and safety in treating coronary heart disease, angina pectoris, obesity, and blood pressure [2-5]. Safflower, as the gentleman medicine of Xuebijing injection, has apparent clinical effects on for sepsis [6]. Danhong injection is a medicinal preparation based on Salviae Miltiorrhizae and Flos Carthami (safflower), was also used in the clinical therapy of cardiovascular and cerebrovascular diseases in China for many years [7].
Reference
- Li, S.; Zhang, B.; Zhang, N. Network target for screening synergistic drug combinations with application to traditional Chinese medicine. BMC Syst Biol 2011, 20, S10 (1-13).
- Zhang, Q.; Peng, J.; Zhang, X. A clinical study of safflower yellow injection in treating coronary heart disease angina pectoris with Xin-blood stagnation syndrome. Chin J Integr Med 2005, 11, 222-225.
- Lu, Q.; Xu, J.; Li, Q.; Wu, W.; Wu, Y.; Xie, J.; et al. Therapeutic efficacy and safety of safflower injection in the treatment of acute coronary syndrome. Evid Based Complement Alternat Med 2021, 16, 6617772 (1-10).
- Ruyvaran, M.; Zamani, A.; Mohamadian, A.; Zarshenas, M. Eftekhari, M.; Pourahmad, S.; et al. Safflower (Carthamus tinctorius) oil could improve abdominal obesity, blood pressure, and insulin resistance in patients with metabolic syndrome: A randomized, double-blind, placebo-controlled clinical trial. J Ethnopharmacol 2022, 10, 114590 (1-44).
- Chen, Y.; Li, M.; Wen, J.; Pan, X.; Deng, Z.; Chen, J.; et al. Pharmacological activities of safflower yellow and its clinical applications. Evid Based Complement Alternat Med 2022, 27, 2108557 (1-24).
- Zhou, W.; Lai, X.; Wang, X.; Yao, X.; Wang, W.; Li, S. Network pharmacology to explore the anti-inflammatory mechanism of Xuebijing in the treatment of sepsis. Phytomedicine 2021, 85, 153543 (1-9).
- Feng, X.; Li, Y.; Wang, Y.; Li, L.; Little, P.; Xu, S.; et al. Danhong injection in cardiovascular and cerebrovascular diseases: Pharmacological actions, molecular mechanisms, and therapeutic potential. Pharmacol Res 2019, 139, 62-75.
Reviewer 2 Report (New Reviewer)
The revised version of the manuscript is appropriate for publication.
Author Response
Thank you very much.
Round 3
Reviewer 1 Report (New Reviewer)
The revision is acceptable.
This manuscript is a resubmission of an earlier submission. The following is a list of the peer review reports and author responses from that submission.
Round 1
Reviewer 1 Report
I have read the paper entitled „Mechanism Repositioning Based on Network Pharmacology: Anti-Inflammatory Effect of Safflower on Myocardial Ischemia Reperfusion Injury” with great interest by Zhao Feng et al.
This study reports on the effects of Safflower in a LAD ligation model of myocardial ischemia-reperfusion (MIR). In a novel approach, the components of Safflower were identified, followed by literature mining to identify potential targets of the active compounds. Finally, efficacy of Safflower was demonstrated by detecting expression changes of specific target genes of MIR and by functional investigations in the model.
Despite the considerable merits of the article, a few questions remain to be answered before publication.
- It should be clarified on what basis the effective dose of Safflower is determined. Three doses were used and the higher dose was more effective. The question logically arises: what would have happened if the dose had been increased further?
- Can there be any unexpected effects of high doses of Safflower, and have any of these effects been observed during the experiments?
- In literature mining, did the level of evidence of the publications included matter? Were they weighted by number of cases? Were clinical studies and animal studies included at the same time?
-The number of animals used in the experiment is not entirely clear. In the Materials and Methods section (4.2.2.), 36 animals are mentioned, divided into 6 groups. I assume that the same animals were not used for TUNEL and TTC staining. By the way, the Fig 2 legend does not include these numbers either.
Minor comments:
UPLC-QTOF-MS/MS is not defined while the much more commonly known quantitative real-time polymerase chain reaction is written out in the abstract.
Correct this sentence: "Thirty-six mice were randomly divided into six groups, including the sham, model, safflower, medium doses, high doses (62.5 mg/kg, 125 344 mg/kg, 250 mg/kg, 4 times/24 h, i.v.), and metoprolol groups (12.5 mg/kg/d, i.p.). "
Author Response
Thank you very much for your comments. Please see the attachment。

Reviewer 2 Report
In this manuscript entitled “Mechanism Repositioning Based on Network Pharmacology: Anti-Inflammatory Effect of Safflower on Myocardial Ischemia Reperfusion Injury”, Feng et al. purposed to discover the mechanisms involved in the cardioprotection induced by Safflower in an in vivo model of myocardial ischemia-reperfusion (MIR) injury. For this purpose, the authors showed that Safflower extract has protective effects, identified the major compounds of the extract, predicted the mechanism involved in the cardioprotection by screening the genes implicated in the Safflower mechanism of action on genes databases of MIR injury and crossing this data with the compounds of the extract, and measured the predicted targets. At the current stage, this study is only incremental. Please, see below the concerns of this reviewer.
Major concerns
There is no novelty in the effects of Safflower on MIR injury, on its major compounds, or in its targets for cardioprotection.
The study methods were not thoroughly described, making it difficult to understand and reproduce fully.
The integrative pharmacology strategy used is already largely known and has already been used in Safflower extract studies.
The studies to confirm the mechanism of action of Safflower extract only evaluated inflammatory markers without a proper investigation.
General comments
The motivation of the study can be further improved in the Introduction section.
The study results need to be clearly described.
It would be helpful for your reader to see a brief methodological description of the study design when introducing its results.
It is unclear why the authors used metoprolol as a positive control.
No data from Sham group is presented from TTC/evans blue staining.
Not the same portion/arThe ea of the heart was used for TTC/evans blue staining, which may bias the analysis since a remote zone from some hearts may have been used.
“Model” group should be named “Control”.
The parasternal long axis is the most appropriate view to analyze the cardiac function in this model.
Please provide better representative images from echocardiographic studies.
Additional cardiac function data should be provided.
There is no evidence that safflower improved the contractile function in a dose-dependent manner.
Some headings in the results section could be merged for conciseness and clarity.
An inadequate sample size was used for TUNEL staining.
It is not clear what means the signals used for statistical differences.
More information should be shown the Fig. 2J. Quantification should also be provided.
The sample size described in the figures does not match the description in the methods section.
No information on when and how the treatment was performed is provided.
Section 4.4. lacks detail about the data entered into the tools used.
The discussion section needs to be improved. In the actual state, the discussion section only describes the results already presented.
Page: 2, Line: 48 – what do the authors mean by “include activation of blood circulation”?
Not all abbreviations are presented on the first time of appearance.
The manuscript lacks a conclusion.
Language revision is needed in some places.
Round 2
Reviewer 2 Report
Despite all the authors' efforts, the manuscript is still incremental and lacks novelty.